# Who is in the sample? An analysis of real and surrogate users as participants in user study research in the information technology fields



Joni Salminen[1,2,3], Soon-gyo Jung[1], Ahmed Kamel[4], Willemien Froneman[5] and Bernard J. Jansen[1]

[1] Qatar Computing Research Institute, Hamad Bin Khalifa University, Doha, Qatar
[2] Turku School of Economics, University of Turku, Turku, Finland
[3] School of Marketing and Communication, University of Vaasa, Vaasa, Finland
[4] Department of Clinical Pharmacy, Cairo University, Cairo, Egypt
[5] Africa Open Institute, Stellenbosch University, Stellenbosch, South Africa

Corresponding author
Joni Salminen,
jsalminen@hbku.edu.qa

## ABSTRACT

**Background:** Constructing a sample of *real users* as participants in user studies is considered by most researchers to be vital for the validity, usefulness, and applicability of research findings. However, how often user studies reported in information technology academic literature sample *real users* or *surrogate users* is unknown. Therefore, it is uncertain whether or not the use of *surrogate users* in place of *real users* is a widespread problem within user study practice.

**Objective:** To determine how often user studies reported in peer-reviewed information technology literature sample *real users* or *surrogate users* as participants.

**Method:** We analyzed 725 user studies reported in 628 peer-reviewed articles published from 2013 through 2021 in 233 unique conference and journal outlets, retrieved from the ACM Digital Library, IEEE Xplore, and Web of Science archives. To study the sample selection choices, we categorized each study as generic (*i.e.*, users are from the general population) or targeted (*i.e.*, users are from a specific subpopulation), and the sampled study participants as *real users* (*i.e.*, from the study population) or *surrogate users* (*i.e.*, other than *real users*).

**Results:** Our analysis of all 725 user studies shows that roughly two-thirds (75.4%) sampled *real users*. However, of the targeted studies, only around half (58.4%) sampled *real users*. Of the targeted studies sampling *surrogate users*, the majority (69.7%) used students, around one-in-four (23.6%) sampled through crowdsourcing, and the remaining 6.7% of studies used researchers or did not specify who the participants were.

**Conclusions:** Key findings are as follows: (a) the state of sampling *real users* in information technology research has substantial room for improvement for targeted studies; (b) researchers often do not explicitly characterize their study participants in adequate detail, which is probably the most disconcerting finding; and (c) suggestions are provided for recruiting real users, which may be challenging for researchers.

**Implications:** The results imply a need for standard guidelines for reporting the types of users sampled for a user study. We provide a template for reporting user study sampling with examples.

## INTRODUCTION

Sampling *real users* for user studies is vital for ensuring a user study's validity, usefulness, and applicability. *Real users* are *study participants that are drawn from the specific target population*. The target population is the entire group of people that a researcher is interested in studying and analyzing, which is often related to a system or phenomenon. Examples of a target population could be the customers of a company, people who own a given device, or the residents of a particular country. However, the target users could also be the general population, meaning any person would be included in the target population. Examples include generic search services, smartphone interactions, or broad social media platforms that target a global user base. Defining the *real users* of a technology raises the interesting concept of "non-users." While there has been a degree of conceptual work on distinguishing between users and non-users (*Augustin et al., 2021*), much of the focus of information technology (IT) user studies has traditionally been on the 'user,' whereas the 'non-user' has been neglected. However, the research reported here tackles the adjacent problem of sampling *real* or *surrogate users* in user studies.

*Real users* are selected from the target population using one or more sampling techniques and become the sample used for the study. The sample is the set of people representing the target population participating in the investigation. Sampling is the process of determining the participants taken from the target population for the study and can use techniques such as a sample frame (*e.g.*, a list of individuals in the target population). The methodology used to sample a larger population depends on the analysis and may include, for example, random, stratified, or systematic sampling techniques. The people who participate in the user study are referred to as participants. So, the terms 'user' and 'participant' are roles that refer to actual people concerning the study. In the 'user' role, people use the technology, and in the 'participant' role, people engage in the study.

Although findings from *real users* may be confounded by sampling issues such as convenience sampling (*Etikan, 2016*), the employment of *real users* from the target population is a critical first step in the user study process. If researchers sample *surrogate users* (*i.e.*, those who are not part of the intended target population of the tested technology) in place of *real users* (*i.e.*, those who belong to the intended target population of the tested technology), they risk gaining feedback and reporting results that do not correspond with actual use cases or that do not reflect the views of the intended users of the technology, thereby conveying erroneous findings to the research community. While using *real users* is not the only step in an appropriate sampling process, it is undoubtedly a necessary first step. Additionally, as presented in the discussion section of the current article, there may be situations where *surrogate users* can provide useful feedback. However, in the main, it seems reasonable that one would choose *real users vs. surrogate users* for most user studies, in most cases.

The type of users sampled is critical because many systems, algorithms, user interfaces, and techniques require feedback from *real users* (*Fotler et al., 2022*) to be adequately evaluated, given that they are designed for specific target populations (*Fischer, Peine & Östlund, 2020*). For example, testing an application to facilitate the workflow of homecare nurses with anonymous crowd workers could yield questionable or even dangerous conclusions (as in harming populations that are dependent on the technology being tested) about user behavior. Concerns about not sampling *real users* but *surrogate users* have caused some publication venues to issue warnings. For example, the *Journal of Advertising Research*, at the time of the study, warns that sampling students or crowd workers in primary studies may lead to a desk rejection:

> *"Authors should clearly articulate the sampling frame and relevant details including response rates and tests for non-response bias. While samples using students or Amazon's Mechanical Turk (MTurk) are appropriate for pilot studies, use in main studies likely will lead to desk rejection. We strongly prefer more generalizable sample populations."*[1]

However, prior work is ambivalent on the actual impact of sampling *real vs. surrogate users* for user studies, with some decrying the practice (*e.g.*, *Chandler & Shapiro, 2016*; *Galloway, 2005*; *Peterson & Merunka, 2014*; *Pickering & Blaszczynski, 2021*) and others stating that it is not a serious problem (*e.g.*, *Hultsch et al., 2002*; *Losada et al., 2021*; *Steelman, Hammer & Limayem, 2014*). While sampling *real users* is considered essential by many researchers for determining a study's validity and practical impact, due to the need for appropriate sampling, it is unclear how frequently user studies in IT (*e.g.*, computer science, information science, and human-computer interaction (HCI)) employ *real users vs. surrogate users*. If researchers frequently employ *surrogate users*, this may indicate a problem with the external validity of the body of research. However, if researchers usually employ *real users* employing appropriate sample techniques to select the study participants, there may not be a widespread issue. The current body of knowledge does not clearly indicate how common the practice of sampling *real vs. surrogate users* is for user studies in the IT fields. Thus, the lack of insight concerning this vital issue is the motivation for our research.

To investigate this issue, we pose the following research questions (RQs):

- **RQ1:** *Do user studies reported in IT literature sample real users or surrogate users as participants?*
- **RQ2:** *If sampling surrogate users, what are the types of participants in these user studies?*
- **RQ3:** *Are there differences in the sampling of real users or surrogate users as participants in the IT fields being studied?*

To address these RQs, we extract information from 725 user studies using an approach similar to that employed in related review work (*Chan et al., 2019*; *Yasen & Jusoh, 2019*). We analyze the results using a mixed-method approach, providing both quantitative and qualitative analyses. Our objective is to better understand the sampling of *real* and *surrogate users* in IT. For this review, a user study is defined as a *methodical evaluation*

---

[1] Journal of Advertising Research Guidelines for Contributors: http://www.journalofadvertisingresearch.com/sites/default/files/Additional_assets/JAR%20Guidelines.pdf.

*using people interacting with IT to assess the performance of a system, and/or people's reactions to the technology or technology outputs.* User studies can take many forms, such as A/B testing with a specific algorithmic technique, system, or various usability evaluations. IT can refer to a complete system, a software application, or a component of an overall information system. A system can refer to a complete system or a component of a system, such as an algorithm, interface, or software output (*e.g.*, a search result listing).

This work presents a systematic literature review (*Snyder, 2019*). A systematic literature review has clearly formulated research questions, identifies relevant studies related to those questions, appraises the quality of the relevant articles, and summarizes the evidence using an explicit methodology (*Khan et al., 2003*). A systematic review usually uses two or more academic databases for article retrieval, applying a well-defined search strategy that is made public and can be replicated by other scholars. Each located article is screened for inclusion/exclusion based on predefined screening criteria, and then analyzed using data extraction (typically a spreadsheet) that leverages a predefined set of criteria that address the research questions. Our systematic review provides an analysis of practice in the state-of-the-art of user studies concerning the single theme of sampling *real* or *surrogate users*. Including 628 articles spanning nearly a decade of research, the number of research articles we analyze covers a wide range of IT fields. The articles are drawn from two major digital libraries and further supplemented with snowball-inspired sampling (for a detailed review of snowballing sampling, see *Wohlin, 2014*) to identify relevant and influential sources from a third digital library. Therefore, we believe the results of our research will provide impactful insights for the IT community.

## Why do real users matter?

Three distinct foundational problems (FP) of not sampling *real users* in user studies are discussed in this section.

### FP1: Validity

Regarding how well findings can be generalized to other situations, user studies that do not employ *real users* face external validity issues (*Cohen, Manion & Morrison, 2017*; *Gundry & Deterding, 2019*). Particularly, if the participants do not belong to the target population (*i.e.*, *surrogate users*), they may lack the motivation, ability, or expertise (*Yesilada, Brajnik & Harper, 2009*) to give valid responses (*Krawczyk, Topolewski & Pallot, 2017*; *Muhib et al., 2001*; *Ritchie et al., 2013*). Although the employment of *real users* does not necessarily address the potential lack of motivation, it does ensure a sufficient degree of domain expertise for external and ecological validity (*i.e.*, how well the results predict behavior in real life).

### FP2: Usefulness

While scientific validity is aimed at the accuracy and precision of results, there is a quintessential question underlying the employment of the research findings: *Are the results useful for researchers and practitioners?* (*Beel & Langer, 2015*; *Brittain, 1975*; *Kosara et al., 2003*). Usefulness—or accuracy—is unlikely to be achieved when using *surrogate users*, as the needs and problems discovered may not match those faced by the intended

users of the technology (*Arkush & Stanton, 1988*). Therefore, regardless of sampling validity, *surrogate users* are unlikely to provide helpful feedback when compared to feedback drawn from *real users*.

### FP3: Applicability

When *real users* are not included in the sample, researchers may miss valuable insights for improving the technology (*Rahi, 2017*; *Schillewaert, Langerak & Duharnel, 1998*; *van Berkel & Kostakos, 2021*). This is related to the issue that the sample may not represent the underlying population that will employ the technology (*Kujala & Kauppinen, 2004*). Since *surrogate users* lack intimate knowledge of a domain, subject matter, or problem space, they cannot provide feedback that would foreseeably lead to new features and functionalities addressing an impactful problem for the targeted population.

## Research gap

Compatible with the aforementioned reasoning, researchers in several fields have raised the need to engage with *real users*. First, many **user-centered design studies** (*Kashfi, Nilsson & Feldt, 2017*) speak of sampling *real users* by adopting the users' *point of view* (*Dourish, 2006*) (p. 542), and *Abras et al. (2004)* advocate the involvement of *real users* in testing designs and prototypes.

Second, researchers in **business studies** (*e.g.*, tourism, marketing, management) have long emphasized "value co-creation" (*Mustak, Jaakkola & Halinen, 2013*), meaning letting customers participate in product development in one way or another. This can be seen as a form of sampling *real users*.

Third, the idea of customer involvement has also been raised in the **IT startup sector**, where influencers (*Blank, 2013*; *Ries, 2011*) have vigorously argued for testing and validating ideas in the real world and "*getting out of the building*" to meet the real customers early on in the product development process. This notion of enabling user or customer participation in the development cycles of products has also been embodied in the developmental concept of the "living lab" (*Almirall & Wareham, 2009*) that many modern startup incubators and research laboratories embody.

Finally, academics in various sectors have warned against the excessive use of students as study participants when performing empirical investigations (*Carver et al., 2004*). The concern is that this may yield non-representative samples of the target population (*Blair & Zinkhan, 2006*), and the sampling of students certainly raises questions of participant motivation, as well as issues of ethics, undue pressure, and fair compensation.

Specifically, one can observe a need to engage with *real users* being presented by researchers from nearly all disciplines:

- From user-centered design, "[optimal designs] *can only truly be achieved by involving real users throughout the design process*." (*Wilkinson & De Angeli, 2014*, p. 627), and "*it is crucial that real users are included as workshop participants. Without real users [one] runs the danger of 'spinning in the air' and simply iterate existing assumptions and prejudices about the context of use*." (*Svanaes & Seland, 2004*, p. 482);

- From service design, "[be cautious of] *generalizing your own personal and subjective experience without cross-checking with real users*" (*Buchenau & Suri, 2000*, p. 432), and "*it is crucial for getting managers outside their narrow organizational view and into the minds and hearts of real customers.*" (*Liedtka & Ogilvie, 2012*, p. 9);

- From business/tourism, "*It is suggested that the testing of new services should be conducted live, with real customers, and in real transactions.*" (*Konu, 2015*, p. 4) and "*prototyping and experimentation produced conversations with real customers, a better source of information than PowerPoint presentations to colleagues in conference rooms.*" (*Liedtka, 2014*, p. 44);

- From entrepreneurship, *lean startups use a 'get out of the building' approach called customer development to test their hypotheses. They go out and ask potential users, purchasers, and partners for feedback on all elements of the business model*" (*Blank, 2013*, p. 1).

From a scientific perspective, sampling users from the targeted population has long been a tenet for valid sampling techniques. After all, using *real users* for a user study makes intuitive sense, and evaluating the use of a system designed for a target population with members of that population is clearly the most sensible approach. Therefore, the issue of whether or not *real users* are being employed in IT research is of paramount importance for the validity (FP1), usefulness (FP2), and applicability (FP3) of user study findings. However, while these merits and concerns associated with sampling *real users* are recognized in theory, there are no guarantees for their realization in practice, as the practical research work often contains multiple sources of distraction. For example, budget requirements, stringent deadlines, and the difficulty of recruiting subject-matter experts incentivize researchers, in a real sense, to seek convenience samples such as *surrogate users*.

These considerations indicate a need for literature analysis (*Torraco, 2005*). Overall, the IT community should be aware of the state of using *real users* in user studies so as to be able to trust the reported findings of such research. Therefore, it is somewhat surprising that previous IT research does not provide a detailed answer to whether user studies use *real users*, and instead, the focus of literature surveys on user studies in the IT fields has mostly been on methods, findings, and data sources.

Table 1 shows that the sample size of reviewed articles in surveys addressing aspects of user study research is often small, and there is a lack of consideration for *real vs. surrogate users* (*Bautista, Lin & Theng, 2016*; *He & King, 2008*; *Lee & Cunningham, 2013*; *Van Velsen et al., 2008*; *Varghese, 2008*). Our research addresses this specific and essential point that is missing in the prior literature. As shown, only one of the user study reviews in the existing literature addresses the use or nonuse of *real users* in the studies. *Kim et al. (2013)* focused on user studies in the smart homes field, finding that, of the fifty-eight articles reviewed, only 20 included user studies with rigorous evaluation, and 12 (20.7%) of the studies used what could be defined in this research as *real users*. This finding hints at concerns for the external validity of user studies in this area.

**Table 1 Previous literature reviews focusing on user studies.** There is a lack of addressing the use of *real users*, and the studies generally review a small number of articles.

| Author | Addresses use of *real user*s | Sample size |
|---|---|---|
| *He & King (2008)* | no | 82 |
| *Varghese (2008)* | no | 101 |
| *Van Velsen et al. (2008)* | no | 63 |
| *Bautista, Lin & Theng (2016)* | no | 10 |
| *Lee & Cunningham (2013)* | no | 198 |
| *Kim et al. (2013)* | yes | 20 |

## SURVEY METHODOLOGY

Implementing a systematic review requires using a framework to synthesize findings from multiple research studies. To address our research questions, a systematic search using the search phrase 'user study' was conducted on the major information technology databases ACM Digital Library and IEEE Xplore, with further snowball-inspired sampling conducted on the Web of Science database to accrue high impact articles published from 1 January 2013 to 31 December 2021. We limited the search to peer-reviewed articles or conference proceedings published within this period, with the full text of the article available and written in English. We explain the process in more detail immediately below.

### Collection of articles

Figure 1 illustrates the literature collection process using the PRISMA flow diagram (*Sarkis-Onofre et al., 2021*). As a systematic review, we found the PRISMA flow diagram and process (*Amin et al., 2018*) to be helpful in presenting and reporting our data collection process.

We first searched the ACM Digital Library (ACMDL) with the phrase 'user study' in the advanced search bar, as the ACMDL is a comprehensive archive of IT research containing arguably the leading HCI conferences in the field, several related conferences, and related leading IT journals that report user study research. The ACMDL also spans a wide assortment of IT fields, and user study research conducted in these fields. We conducted the search on 22 March 2022, with a date limit of 2013 through 2021, for full-text articles. The search yielded 11,653 results published until 2021, which from a practical perspective, were too many to classify manually. In particular, we needed a manageable sample size for manual classification because identifying whether a study uses *real* or *surrogate users* requires human insight and cannot be determined automatically. Therefore, we narrowed the results to articles with 'user study' included in the article's keywords. This search yielded 658 unique articles—a number that was deemed manageable for manual classification. The articles were then downloaded from ACMDL for manual classification. The specific search query used was:

"query": {Keyword:("user study")}

"filter": {Publisher: Association for Computing Machinery, Publication Date: (01/01/2013 TO 12/31/2021), ACM Content: DL}

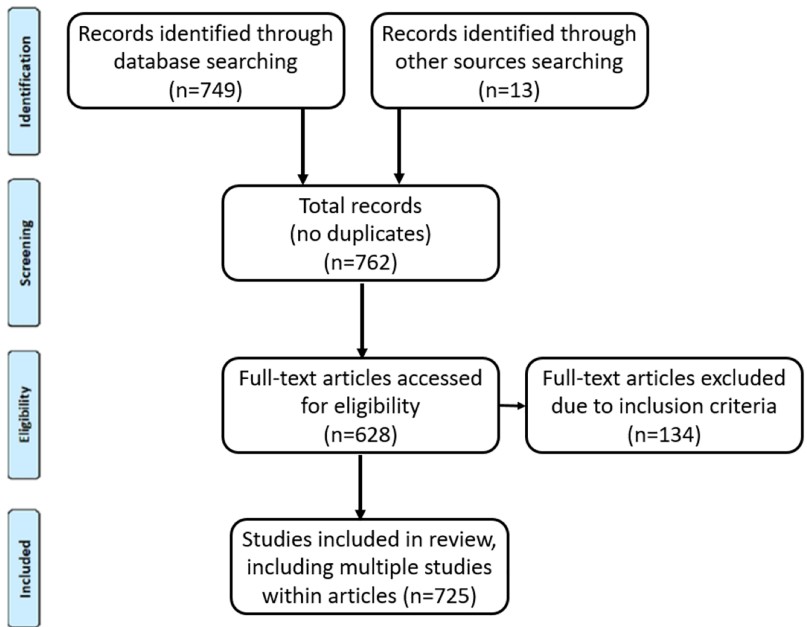

**Figure 1 PRISMA flow diagram of the literature collection process, with a full explanation provided in the text.**

In order to mitigate possible source bias by collecting user studies from just the ACMDL database (despite it being a diverse archive), we then searched the IEEE Xplore digital library with the phrase 'user study' as the keyword in the advanced search feature, as IEEE Xplore is another comprehensive archive of IT research. IEEE Xplore also contains a wide assortment of IT fields and user study research in these fields. This search yielded 2,576 results published from 2013 to 2021, which were again too many to manually classify, and from a review of the abstracts, most did not contain a user study. Since using the keyword field returned too many false positives, we instead searched for articles with 'user study' included in the article's title for conferences and journals published in IEEE Xplore. This search yielded 91 unique articles—a number that was deemed manageable for manual classification. The articles were then downloaded from IEEE Xplore for manual classification. The specific query applied was:

("Document Title":"user study")

Filters Applied: "Conferences Journals" 2013–2021

Finally, using a snowball-inspired sampling approach, we identified a further 13 highly cited user study articles in the Web of Science (WoS) database from various journal outlets by using the 'highly cited' feature in the WoS interface and the search phrase "user study." As we wished to keep all of the articles in the study within the same period, we again limited the selection to articles from 2013 to 2021. Snowballing using citations of sources employed in the returned articles is a common technique in systemic reviews (*Wohlin, 2014*). Our approach of using highly cited articles from the WoS is similar, but rather than using the reference lists of the selected articles, we leverage this additional database to identify relevant and highly cited articles. As mentioned by *Wohlin (2014)*, there is always a

need to identify relevant literature, and so different techniques should be employed to maximize the chances of discovery. The use of the WoS feature of filtering for highly cited articles is certainly more efficient than the traditional method of picking cited articles from the reference list of articles contained in the initial dataset. However, further experimentation would be needed to determine the pros and cons of the two techniques of traditional snowball sampling or snowball-inspired sampling.

The combined searches on three archival databases resulted in 762 articles. Our focused search criteria and snowball-inspired sampling resulted in no duplicate articles needing to be removed. We manually screened the articles by inspecting whether each retrieved article actually reported a user study. To determine studies to be included in this review, the following inclusion/exclusion criteria were used: (a) The article had to report an actual empirical user study involving interaction with an IT system. So, for example, if the article was a literature review, it was excluded. The article was also excluded if it reported only a survey (*i.e.*, without an IT interaction). (b) The article had to report the involvement of study participants. Articles that only discussed algorithms or other technical aspects were excluded. Applying the inclusion/exclusion criteria *via* manual inspection led to 134 articles (17.6%) that did not contain an empirical user study of an IT system being discarded.

Accordingly, the number of articles with actual user studies was 628 (82.4%). Of these, 10 (1.6%) articles contained three separate studies, and 77 (12.3%) articles contained two separate user studies. Combined with the 541 (86.1%) articles containing one user study, this resulted in 725 unique user studies for our evaluation ($541 + (10 \times 3) + (77 \times 2)$). The 725 studies were methodically reviewed in order to extract the relevant data. To organize the findings in each article, the type of IT study, the type of users, and the IT domain were used as the analysis framework. For each selected article, the pertinent information was recorded in a spreadsheet (see Appendix 1 for details of each article).

## Data extraction

We extracted a range of information from the articles (see Table 2).

To classify whether an article samples *real users* or not, we define the following concepts:

- **Real user** is *a participant in the user study that is likely to be from the target population that uses the technology that the research article is presenting.*
- **Surrogate user** is *a participant in the user study that is not likely to be from the target population that uses the technology that the research article is presenting (i.e., a participant that is not a real user).*

In our definitions, by 'likely', we mean that the participants would/would not probabilistically belong to the target population. We acknowledge that the distinction between *real* and *surrogate* is not always clear and distinct. For example, student participants would be *surrogate users* if used to evaluate an IT system designed for experienced medical professionals, but medical student participants that are about to graduate might qualify as *real users*. Another example would be user studies of virtual

**Table 2 User study information extracted from the articles.** [A] denotes automatic retrieval using ACMDL, IEEE Xplore, and WoS export function, while [M] denotes manual extraction researchers.

| Attribute | Definition |
| --- | --- |
| **Article focus** | |
| Title [A] | Article title |
| Year [A] | Year of publication |
| Domain [M] | Domain of the study (*e.g.*, mobile computing) |
| Outlet [A] | Venue that published article |
| Type of publication [M] | Type of publication venue (Journal/Conference). |
| **User studies reported within the article focus** | |
| Targeting [M] | Type of targeting (Generic – no defined user group/Targeted – defined user population) |
| No. of participants [M] | The number of participants in the user study. |
| Type of users [M] | Participants in the user study (*e.g.*, *real user*s or *surrogate users*) |
| Type of Surrogate User [M] | If *surrogate user*s, then what type (*e.g.*, students, crowdworkers, researchers, not mentioned) |

reality gaming. Technically, these technologies could target everyone. However, when testing a new virtual reality game, some interest or expertise in gaming would seem appropriate. Another edge case example would be mobile technology. Most mobile phone studies would be generic. However, some such studies focused on specific domains, such as virtual reality gaming or tourism in a particular location. So, this required a manual evaluation of each study in order to make the determination.

## Coding principles and categories

Manual coding was carried out by reading the articles and saving the information in a spreadsheet. If multiple studies were part of the same article, they were analyzed separately (*e.g.*, if an article had one qualitative study with eleven participants and one quantitative study with 201 participants, the two studies were recorded in separate rows).

## Type of user study

Some technologies that are tested do not have a specific target group in mind but are rather designed for a general population. Therefore, we devised a study classification of *Targeting* (see Table 2), with the values *Generic* and *Targeted*. There is no predefined target population for generic technologies, so every person is potentially a user of the technology. For targeted technologies, however, there is a defined user population. We acknowledge that there may be situations where the difference between *Generic* and *Targeted* technology is fuzzy or that generic technology could later be employed with a targeted population.

Generic technology examples could be testing a search result page, a mobile phone UI, or something else intended for the general public. In contrast, targeted technologies have a specific target group, such as nurses, hikers, graphic designers, *etc*. A mobile application for nurses only targets nurses, so students (unless they are nursing students) are not *real users*. Targeting can be based on factors such as demographics (*e.g.*, "the needs of the elderly": *Diepold et al., 2017*), physical condition (*e.g.*, "visually impaired people": *Alkhanifer & Ludi, 2014*), or behavior (*e.g.*, "the motion sickness experienced by passengers who read

using tablet computers or phones": *Hanau & Popescu, 2017*). Often, targeted technologies require specific skills that not everyone would have (*e.g.*, professional skills). We generally consider the Internet, email, online search engines, and mobile phones to be standard technologies available to the general public in most parts of the world. For generic technologies, because everyone is a potential user, all users in generic technology studies were automatically coded as *real users*.

## Type of users employed

Even with the employed definitions, determining *real users* remains partly subjective. Overall, the process for determining *real users* involved creating the operational definition of *real users*, internalizing its meaning, and then using the definition to manually classify the reviewed studies in the selected articles. Most user study research articles do not explicitly mention *real users*, and this observation is evident from reading the articles in our sample. Therefore, we needed to infer this from reading the article. The applied user types are (a) *real users* and (b) *surrogate users* (corresponding to the definitions provided in the earlier section).

Moreover, studies use different expressions to describe users—such as volunteers, participants, or stakeholders—and the population focus of the technology is not always clearly presented, nor the targeted population explicitly defined. For these reasons, human judgment is needed to determine if the study uses *real users*. Examples of *real users* include hikers (*Posti, Schöning & Häkkilä, 2014*), housekeeping staff (*Doke & Joshi, 2015*), and nurses and clinicians (*Müller, 2017*), with the qualitative analysis providing more examples.

For *surrogate users*, we further categorized them into four groups:

- **students**—people who is studying at a school or college.
- **crowdworkers**—people recruited *via* a crowdsourcing platform or *via* means such as listserv or social media
- **researchers**—people conducting academic or scientific research, or a faculty or staff member at an educational institution
- **not mentioned**—people who was a participant in a user study, but their categorization was not mentioned in the article

For a small number of studies, there is a mix of participants, and we choose the largest group as the demonstrative group. In nearly all of the studies with mixed participants, they involved students together with researchers and/or academic staff. For example:

> "*We recruited a total of 20 participants […], with ages ranging from 22 to 58 years (M = 31.65). Participants were recruited from the university students and staff and personal contacts of the authors. Nineteen participants had computer gaming experience, and only nine participants had experience with VR.*" (*Chen et al., 2017*) (p. 112)

In this example, it appears that most participants are students, who are not *real users* of the technology, so we choose *surrogate users*, specifically students.

**Table 3 IT fields of user studies.** The taxonomy presents the fields of user study articles based on open coding using *real* and *surrogate users*. The bolded row shows the mid-point (the average).

| Domains | Real users | % | Surrogate users | % | Total |
|---|---|---|---|---|---|
| Security & privacy | 39 | 97.5% | 1 | 2.5% | 40 |
| Wearable technology | 28 | 93.3% | 2 | 6.7% | 30 |
| Input devices & technologies | 51 | 91.1% | 5 | 8.9% | 56 |
| Recommender systems | 26 | 89.7% | 3 | 10.3% | 29 |
| Mobile & ubiquitous computing | 34 | 85.0% | 6 | 15.0% | 40 |
| e-Health & accessibility | 55 | 84.6% | 10 | 15.4% | 65 |
| Online content & social media | 20 | 83.3% | 4 | 16.7% | 24 |
| Games | 15 | 83.3% | 3 | 16.7% | 18 |
| Government & non-profit | 8 | 80.0% | 2 | 20.0% | 10 |
| Driving & transportation | 19 | 76.0% | 6 | 24.0% | 25 |
| Collaborative work & remote experiences | 29 | 70.7% | 12 | 29.3% | 41 |
| **Mid-point (Average)** | | **76.1%** | | **23.9%** | |
| Business applications | 32 | 69.6% | 14 | 30.4% | 46 |
| 3D models & interfaces | 13 | 68.4% | 6 | 31.6% | 19 |
| Software development | 19 | 67.9% | 9 | 32.1% | 28 |
| Information processing & search | 63 | 67.7% | 30 | 32.3% | 93 |
| Robotics & artificial intelligence | 19 | 63.3% | 11 | 36.7% | 30 |
| Digital analytics & visualization | 27 | 61.4% | 17 | 38.6% | 44 |
| Virtual & augmented reality | 38 | 58.5% | 27 | 41.5% | 65 |
| e-Learning & education | 12 | 54.5% | 10 | 45.5% | 22 |
| Grand total | 547 | 75.4% | 178 | 24.6% | 725 |

## Fields

We defined IT fields based on our open coding, using standard domain terms as labels (see Table 3). An IT field refers to a defined sphere of knowledge.

The categories for IT fields were created using an open coding technique (*Glaser & Strauss, 1967*), which is appropriate when there is a lack of suitable pre-existing taxonomies (*Hruschka et al., 2004*). We considered using the *ACM Computing Classification System*, but we found it inadequate for distinguishing between the fields. For example, the system lacks explicit classes for *Mobile Computing* and *Recommender Systems*. Therefore, we developed our own taxonomy for user studies. This taxonomy was developed inductively by analyzing the types of user studies in the sample. Even though not all conceivable fields may be represented, the sample included hundreds of studies across different fields, and the resulting taxonomy encompasses an extensive array of fields where user studies are conducted. The 19 IT fields used to classify the 725 user studies are as follows:

- 3D Models and Interfaces
- Business Applications

- Collaborative Work and Remote Experiences
- Digital Analytics and Visualization
- Driving and Transportation
- e-Health and Accessibility
- e-Learning and Education
- Games
- Government and Non-Profit
- Information Processing and Search
- Input Devices and Technologies
- Mobile and Ubiquitous Computing
- Online Content and Social Media
- Recommender Systems
- Robotics and Artificial Intelligence
- Security and Privacy
- Software Development
- Virtual and Augmented Reality
- Wearable Technology

*Business Applications* include industrial and consumer-market applications (*e.g.*, electric power grids: *Romero-Gómez & Diez, 2016*), aeronautics (*Rice et al., 2016*), supermarkets (*Kalnikaitė, Bird & Rogers, 2013*), banking (*Panjwani et al., 2013*), shipping (*Vartiainen, Ralph & Björndal, 2013*), smart homes, and others with applicability within an industry niche, although not necessarily comprising a full system. *Collaborative Work and Remote Experiences* include group- and teamwork-related studies, as well as crowdsourcing (*Kairam & Heer, 2016*). *Digital Analytics and Visualization* is similar to information processing in that it deals with information. However, this category focuses explicitly on analytical tasks (*Vimalkumar et al., 2021*), such as the presentation of numbers, ontologies (*Zhang et al., 2015*), tables, metrics (*Miniukovich & De Angeli, 2015*), and summarizations (*Rudinac & Worring, 2014*).

*Driving*, *Digital Analytics*, and *Wearable Technology* are separated into their own categories due to a large number of respective studies, although they could be considered as directed business verticals. *Driving and Transportation* include automated vehicles (*Walch et al., 2015*), along with pedestrians (*Bertel et al., 2017*). *Information Processing and Search* deals with various cognitive aspects relating to users (*Song, Liu & Zhang, 2021*). This category includes search, email search (*Kim et al., 2017*), cognitive strategies (*Raptis, Fidas & Avouris, 2017*), and website usability (*Alhadreti & Mayhew, 2017*). *Input Devices and Technologies* include studies focusing on immediate physical contact with the end users, for example, *via* tactile displays (*Chu et al., 2017*) and voice-controlled systems (*Kiseleva et al., 2016*). *Security and Privacy* include studies on authentication (*Winkler et al., 2015*), passwords (*Haque, Wright & Scielzo, 2013*), and so on. *Wearable Technology* includes

smart glasses (*Zhao et al., 2017*), smart watches (*Dibia et al., 2015*), and smart handbags (*Pakanen et al., 2016*).

When a study has characteristics from more than one category, *e.g.*, mobile computing and search, we choose the more dominant one. For example, mobile computing can be a context when the study is really focused on understanding the information processing of the searchers (*Lagun, McMahon & Navalpakkam, 2016*).

The material was coded by a primary researcher who had participated in drafting the coding definitions and was, therefore, highly familiar with their purpose. The researcher reviewed each study carefully to identify whether the study was testing targeted or generic technology and then, for targeted technologies, whether the study participants were real or surrogate users. The consistency of the coding was validated *via* an inter-rater reliability test. A secondary researcher independently coded a sample of 50 randomly selected articles that the primary researcher had already coded. The agreement between these two coders was then calculated using inter-rater reliability metrics, Cohen's Kappa (K), and Interclass correlation (ICC). The obtained scores (ICC = 0.96, K = 0.879, $p < 0.01$) indicate near-perfect agreement (*Gisev, Bell & Chen, 2013*). The number of disagreements nearly exclusively concerned whether a system was designed for a general population (for which students or general crowdworkers might be considered to be *real users*), or a targeted population (for which students or general crowdworkers might be considered as *surrogate users*). Once the type of technology employed in the user study was identified (generic or targeted), the classification of user type was notably easier, as all generic studies employed *real users* by default.

The results of our coding of the 725 studies are included as Supplemental Material for this article.

## RESULTS

After coding the data for analysis, 725 actual user studies were found in the screened 628 articles. These user studies vary by sample size (average = 308; max = 113,682; min = 1; std = 442; median = 23). The most common publication venue was conferences, with 654 (90.2%) user studies published in articles from conferences, and 71 (9.8%) published in journal articles.

Concerning the screened articles, 541 (86.1%) of the articles were from the ACM Digital Archive, 74 (11.8%) from IEEE Xplore, and 13 (2.1%) from WoS. There were 233 unique venues where the 628 unique articles were published. The distribution of articles reporting on user studies presented in this research was generally consistent by year (2013: $n = 86$, 13.7%; 2014: $n = 88$, 14.0%; 2015: $n = 85$, 13.5%; 2016: $n = 96$, 15.3%; 2017: $n = 134$, 21.3%; 2018: $n = 47$, 7.5%; 2019: $n = 5$, 0.8%; 2020: $n = 11$, 1.8%; 2021: $n = 76$, 12.1%), although with dips in 2019 and 2020.

We now address our research questions.

### RQ1: Do user studies reported in the IT literature sample real users or surrogate users as participants?

Of the 725 user studies, 547 (75.4%) employed *real users*, and 178 (24.6%) employed *surrogate users*. However, this includes the 303 (42.8%) user studies that were generic, meaning that any participant would be a *real user*. There were 422 (58.2%) targeted studies in the 725 user studies. Examining the targeted studies, 244 (57.8%) employed *real users* as participants, and 178 (42.2%) employed *surrogate users* as participants.

So, nearly half of the user studies in the IT literature that we analyzed concern generic technologies for which any participant is potentially a *real user*, and just over half of the targeted technology studies sample *real users* as participants.

### RQ2: If sampling surrogate users, what are the types of participants in these user studies?

Examining the 422 targeted user studies, 178 (42.2%) sampled *surrogate users* as participants. Of these, 124 (69.7%) studies sampled students, and 42 (23.6%) studies sampled crowdworkers. Seven (3.9%) studies did not disclose the participant type, and five (2.8%) studies sampled researchers as participants.

So, about seven times out of 10, user studies in IT fields employing *surrogate users* use students as participants.

### RQ3: Are there differences in the sampling of real users or surrogate users as participants by IT field being studied?

We now examine the use of *real* or *surrogate users* by IT field, with the results shown in Table 3.

As shown in Table 3, *Security & Privacy* has the highest occurrence of studies with *real users* (97.5%), with the fields of *Wearable Technology*, *Input Devices & Technologies*, and *Recommender Systems* also at 90% or higher for the sampling of *real users* as participants.

At the other end of the spectrum, *e-Learning & Education* has the lowest sampling of *real users* (54.5%), with *Virtual & Augmented Reality* only slightly higher (58.5%).

Given the mid-point of the sampling of *real users* (see Table 3) at 76.1%, there are apparent differences by IT field. A Chi-Square Goodness of Fit Test was performed to determine whether the proportion of studies with *real users* was equal among the IT fields. The test showed that the proportions did differ by the sampling of *real users*, $X^2(18, 547) = 142.61$, $p < 0.0001$. So, there is a statistically significant difference in the sampling of *real users* by IT field.

Figure 2 visualizes the percentage of *real users* and *surrogate users in samples* across the various fields.

*Security and Privacy* had the highest proportion of *real users* ($n = 39$, 97.5%) (see Table 3). *Wearable Technology* had the second-highest proportion ($n = 28$, 93.3%), and *Input Devices & Technologies* ($n = 51$, 91.1%) was the third-highest. e-Learning & Education had the highest proportion of *surrogate users* ($n = 10$, 45.5%), while *Virtual & Augmented Reality* had the second-highest proportion ($n = 27$, 41.5%), and *Digital*

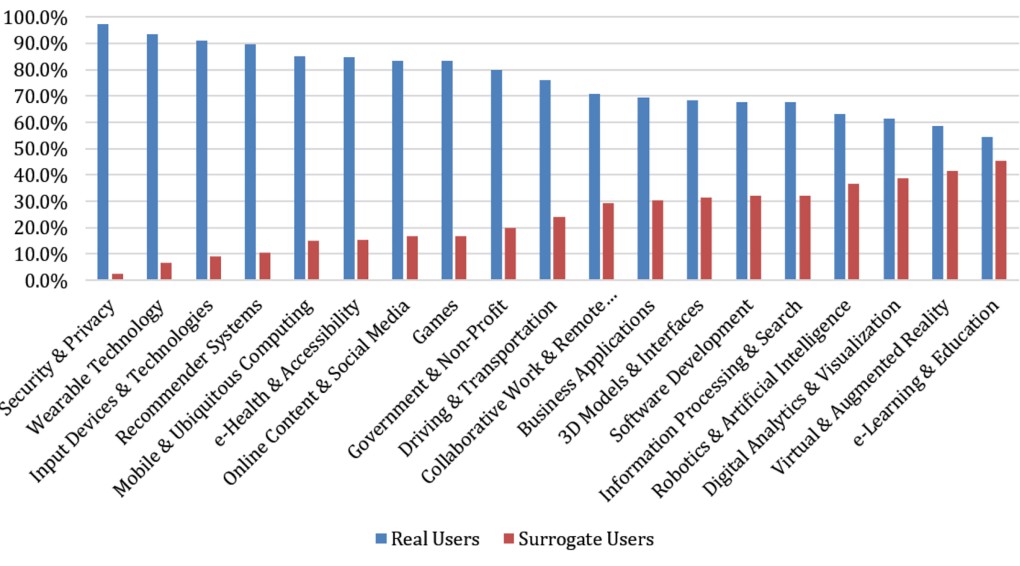

**Figure 2 *Real vs. surrogate users* across IT fields.**

*Analytics & Visualization* (*n* = 17, 38.6%) was the third-highest. As shown in Fig. 2, most user studies in all domains employed *real users*.

The above results concerning the use of *real users* by domain bring into question the sampling of targeted and generic user studies in these fields, which we report in Table 4.

As shown in Table 4, *Recommender System*s had the lowest percentage of targeted user studies (13.8%), with *Security & Privacy* only slightly higher (17.5%). Conversely, *e-Learning & Education* had the highest percentage of target studies (90.9%), closely followed by *Government & Non-Profit* (90.0%).

When comparing the findings presented in Table 3 with those in Table 4, there appears to be a correlation between conducting generic studies and sampling *real users* (or conducting targeted studies and sampling *surrogate users*). A Spearman's rank correlation was computed to assess the relationship between generic studies and the use of *real users*, and the results showed a positive correlation between the two variables, r(723) = 0.616, *p* < 0.05, indicating a moderate positive relationship.

## DISCUSSION

### General discussion about the findings

Returning to our research questions, 75.4% of the user studies in our dataset employ *real users* rather than *surrogate users* as participants (RQ1). However, this percentage includes generic user studies, meaning any participant would be from the targeted population. When examining only user studies of targeted technology, just over half of studies (58.4%) sample *real users*, indicating substantial room for improvement in this regard.

Of the targeted studies employing *surrogate users* (RQ2), the most common type of non-real participant in these user studies is far and away 'students' (accounting for 69.7% of the studies from this research). The use (and perhaps overuse) of students in academic peer review research and the impact on the reported findings is an area for future

**Table 4 Fields of user studies.** The taxonomy presents the IT fields of user study articles based on open coding using targeted or generic user studies.

| Domains | Targeted | % | Generic | % | Total |
|---|---|---|---|---|---|
| Recommender systems | 4 | 13.8% | 25 | 86% | 29 |
| Security & privacy | 7 | 17.5% | 33 | 83% | 40 |
| Input devices & technologies | 18 | 32.1% | 38 | 68% | 56 |
| Online content & social media | 9 | 37.5% | 15 | 63% | 24 |
| Wearable technology | 12 | 40.0% | 18 | 60% | 30 |
| Mobile & ubiquitous computing | 19 | 47.5% | 21 | 53% | 40 |
| Information processing & search | 46 | 49.5% | 47 | 51% | 93 |
| Collaborative work & remote experiences | 21 | 51.2% | 20 | 49% | 41 |
| Virtual & augmented reality | 39 | 60.0% | 26 | 40% | 65 |
| 3D models & interfaces | 12 | 63.2% | 7 | 37% | 19 |
| Games | 12 | 66.7% | 6 | 33% | 18 |
| Digital analytics & visualization | 30 | 68.2% | 14 | 32% | 44 |
| Business applications | 36 | 78.3% | 10 | 22% | 46 |
| Robotics & artificial intelligence | 25 | 83.3% | 5 | 17% | 30 |
| e-Health & accessibility | 56 | 86.2% | 9 | 14% | 65 |
| Driving & transportation | 22 | 88.0% | 3 | 12% | 25 |
| Software development | 25 | 89.3% | 3 | 11% | 28 |
| Government & non-profit | 9 | 90.0% | 1 | 10% | 10 |
| e-Learning & education | 20 | 90.9% | 2 | 9% | 22 |
| Grand total | 422 | 58.2% | 303 | 42% | 725 |

research. Notably, results from studies using students as participants (perhaps in exchange for extra credit in a course) may suffer from 'student bias' (*i.e.*, a lack of motivation where students do the bare minimum to get the extra credit). Prior studies have also raised issues with the use/overuse of students as study participants (*Ashraf & Merunka, 2016*; *Hanel & Vione, 2016*; *Johansen, 2022*).

Concerning differences in the sampling of *real users* as participants in the IT field (RQ3), there are statistically significant differences by field, with some fields nearly always sampling *real users*. All of the fields examined in this research had studies that employed *surrogate users*, although most of the user studies in every field employed *real users*, which is a positive signal. However, many of these user studies were generic studies. Again, the sampling of *surrogate users* in targeted studies was substantially higher (41.6% of the user studies reported here), which is not a positive sign and poses questions about the findings' validity, applicability, and usefulness. There was also a moderately positive, statistically significant correlation between the testing of generic technologies and the sampling of *real users*.

The use of *surrogate users* appears less frequently in fields where user studies are more formally controlled by mechanisms such as using Institutional Review Boards (IRBs) like as *e-Health and Accessibility*, and is more frequent in fields where there is less external

control, such as *e-Learning & Education*. IRBs, or related processes, for example, may be exempt or expedited in these domains, but overall, the fields can be grouped based on their tendency to employ *real users*:

- **High prevalence of sampling *real users* (*i.e.*, well above the mean):** *Security & Privacy*, *Wearable Technology*, and *Input Devices & Technologies*
- **Moderate prevalence of sampling *real users* (*i.e.*, within a standard deviation of the mean):** *Driving & Transportation* and *Collaborative Work & Remote Experience*
- **Low prevalence of sampling *real users* (*i.e.*, well below the mean):** *Virtual & Augmented Reality* and *e-Learning & Education*

### Research novelty

In terms of novelty, our study addresses three key gaps in the previous surveys of user study research:

**Not Focused on Types of Users**: The focus of many of those surveys was on other aspects of the users, such as the methods employed (*Van Velsen et al., 2008*), rather than *who* the users participating in the studies actually were. In this research, we specifically focused on finding out if user studies involved *real users* or *surrogate users*, where *surrogate users* were recruited from, and if the use of *real* or *surrogate users* varied by IT field.

**Small Set of Articles**: The previous literature reviews analyzed small samples, usually tens rather than hundreds of studies (*Bautista, Lin & Theng, 2016*; *Kim et al., 2013*). The sample we analyzed here—725 user studies—is substantially larger than the size of the largest previous review.

**Single Domain**: While the earlier surveys tended to focus on one context or domain such as software systems (*Varghese, 2008*), music (*Lee & Cunningham, 2013*), or social television (*Bautista, Lin & Theng, 2016*), our sample covers a variety of IT fields—19 to be specific—, thus providing a broader perspective on IT user studies than earlier surveys.

### Is there a systematic flaw in user studies?

In terms of the findings of this survey, most IT user studies (75.4%) sample *real users*. However, this includes many studies that are focused on generic technologies that are testing technologies for all user segments. We stress that the use of *surrogate users* can be seen as problematic only when testing specific technologies and not when testing generic technologies (unless specific demographic or behavioral attributes are being evaluated). We also acknowledge that there can be cases where *surrogate users* can provide beneficial feedback or findings, and *surrogate users* might be appropriate in some cases, such as pilot or exploratory user studies.

Of the targeted technology studies, only 58.4% use *real users*, which we feel is a low percentage, and perhaps indicates a problem in the external validity of the findings reported in some of these studies. At the very least, the findings reported in these studies should be taken with a 'grain of salt' (*i.e.*, viewed with skepticism). This percentage is also substantially higher than that reported in *Kim et al. (2013)* analysis of 58 articles, finding that only 12 (20.7%) of the studies sampled what could be defined as *real users*. In the

targeted technology studies that sample *surrogate users*, 69.7% use students, and 23.6% use crowdworkers, with the remaining employing faculty/staff, or not even mentioning the make-up of the sample. These user types are considered as surrogates for *real users* when they are not from the target population of the technology developed, and the impact of the high use of *surrogate users* for targeted technologies is an exciting area for future research.

 At some level, the use of *surrogate users* is understandable for many researchers, as the availability of *real users* can be limited, recruitment of *real users* in sufficient numbers can be quite challenging, and the payment of *real users* in substantial numbers can be somewhat expensive. However, there are methods to address or at least mitigate these hurdles. Based primarily on the experience of the authors, some of these techniques are:

- **Crowdworker Platforms**: Unlike MTurk, which at the time of this study provides limited sample selection features, some other crowdworker platforms offer rather sophisticated sampling methods and access to populations of real users from various domains, demographics, and experiences, mitigating access issues to real users and offering the payment of *real users* at a reasonable cost. Additionally, some of these platforms offer advance quality control features, although, of course, there are both pros and cons of using these crowdworker platforms (*Peer et al., 2022*; *Salminen, Jung & Jansen, 2021*).

- **Partnering with Commercial Companies, Non-profits, and Governmental Organizations**: Paying *real users*, especially in some specialty domains, can be rather expensive. A technique to help mitigate the cost of *real users* for user studies is to partner with a commercial entity (or non-profit or government agency) to provide participants and conduct the user study at the workplace to minimize disruptions to the employees' work schedules. The authors find the offering of a 'thank you' such as a gift card to be a nice touch. Others have also called for closer collaborations by academia with these other forms of organizations (*Lutchen, 2018*; *Mullin, 2021*; *Turin et al., 2022*).

- **Mix of *Real* and *Surrogate Users***: Some studies may require a substantial number of real users that may be unrealistic to recruit or pose a prohibitive expense. A workaround can be a mix of *real users* and *surrogate users*; basically, running two user studies – one with a smaller sample of *real users* and one with a larger sample of *surrogate users*, comparing the results between the two samples. For example, say you need 200 participants to achieve some effect size, and recruiting this number of *real users* is unrealistic, given the availability of *real users* or cost. One technique could be to run user study one with 200 *surrogate users* that are easier to recruit or less expensive. Then, conduct user study two with, say, 30 *real users* (a much more manageable number) as an external validity check. However, this approach does come with the additional cost of executing more than one user study.

 The practicalities of the research process (*i.e.*, the constraints in which researchers design their studies and sampling strategies) further complicate the picture. Namely, one can argue that lab-based experiments sacrifice external validity and realism for internal validity and control (see Fig. 3). Such experiments often use convenience samples,

| External validity / Internal validity | High | Low |
|---|---|---|
| **High** | (Real users) <br> **High Internal and High External Validity** | Surrogate users <br> **High Internal and Low External Validity** |
| **Low** | Real users <br> **Low Internal and High External Validity** | - <br> **Low Internal and Low External Validity** |

**Figure 3 External and internal validity trade-off.** It is possible to obtain both a high external and internal validity using a sample of *real users*, as indicated by the green cell.

performing contrived tasks in artificial environments. Thus, sampling *surrogate users* makes sense when maximizing external validity is not the aim of the study. In contrast, other kinds of studies (*e.g.*, naturalistic workplace studies) may sacrifice internal validity to improve external validity. When the study's purpose is to maximize external validity (as typically is the case for user studies), sampling *real users* is appropriate. As shown in Fig. 3, conventionally, some researchers might think that in order to obtain a high internal validity (control), they need to recruit *surrogate users*; and in contrast, using sampling techniques that maximize external validity would result in lower internal validity. This thinking is illustrated by the grey cells in the figure. However, said trade-off imposes a logical fallacy, as it is possible to obtain both a high external and internal validity using a sample of *real users*, as indicated by the green cell. Naturally, using real users is not the only factor for establishing external validity, with appropriate sampling being an example.

## How are real users used?

The analysis verified that the *real users* in the sampled studies conformed to our working definition of a participant in a user study being likely to be from the target population that uses the technology that the research article presents.

There was a variance in the level of detail provided within the articles on the users; some studies were thorough in describing their users, while others lacked details such as demographics and the means of recruiting the participants. Moreover, some studies provided no details of the participants, or their details were obfuscated.

The *real users* included various groups such as domain experts, stakeholders, members of the general public, and targeted members of the public like children, income groups, visually impaired users, or communities of gamers, illustrating the diversity of contexts where user studies are carried out. Also, in most cases, whether generic or targeted, the participants were selected *via* convenience sampling, although some crowdworker platforms (*e.g.*, Prolific) allowed for stratified sampling. The high use of convenience sampling for recruiting participants in user studies is another area of concern and prompts an area for future research to find a way to incorporate more rigorous sampling methods.

## Contribution to the research community

To ensure the validity of user feedback, we encourage researchers to test their technologies using a "3 Reals" approach: (1) *real users*, (2) *real use cases*, and (3) *real contexts*.

Although these points are not novel in the IT community (for example, see *Dourish, 2006*; *Greenberg & Buxton, 2008*), a repetition of these points seems to be in order. Our contribution is that we stress the point of employing *real users specifically for studies testing targeted technologies*, as researchers do not always explicitly describe their user study sample. The conceptual division between "generic" and "targeted" technologies is necessary in order to judge user study samples appropriately.

Along with this division, IT publications (*i.e.*, journals and conferences publishing user studies) should make *real users* a requirement for user study research that deals with targeted technologies or require justification for why *real users* are not used as participants. To achieve this, a practical solution would be to require, upon submission, an explicit statement from researchers on (a) whether the study tests targeted or generic technology, and (b) whether it employs *real users* or *surrogate users* for the tested technology. These actions would increase the level of transparency in reporting the outcomes of user studies and instill a sense of confidence in the obtained results. They would also make the authors consider whether their technology requires *real users* to achieve validity and usefulness. This 'self-questioning' is worthwhile, as we believe there may be a lack of awareness, along with confusion, as to whom researchers should recruit for their user studies.

A concern is that the IT community often implicitly *assumes* the use of *real users*, instead of requiring an explicit statement of this from the researchers. We suggest making this explicit in each user study by reporting it in a manner similar to an IRB inspection or related processes. So, while IRB is specifically concerned with the ethical aspects of user study research (*Benke et al., 2020*), there is still a gap when considering the validity aspects of targeted technologies user studies.

However, we note that the problem is not only one of validity (*i.e.*, whether the results represent those obtained in real-world circumstances) but also one of *value* (*i.e.*, developing the technology with and for the users). Currently, most authors do not explicitly mention if they used *real users* or not, which forces the readers of these articles to make this conclusion for themselves. For those who lack the sophisticated skills to read academic research papers, this might hinder correctly interpreting the findings.

## Practical implications

We suggest an explicit user study statement for IT user studies, with the following template being provided for researchers and publication venues to report the sampling of users in user studies:

> **"This study tests [NAME OF TECHNOLOGY BEING TESTED] that is classified as a [TARGETED/GENERIC TECHNOLOGY]". To test this technology, we conducted the study in a [WORKPLACE/LABORATORY/ONLINE] environment. For the sample participants, we use [REAL USERS/SURROGATE USERS, who are**

**CROWDWORKERS/STUDENTS/RESEARCHERS/OTHER]. The sampling technique employed was [RANDOM/SYSTEMIC/CONVENIENCE/CLUSTER/ STRATIFIED/OTHER].**

For example, a user study focusing on testing an interactive algorithmically-generated persona system using real users could be reported as follows: *The study tests an algorithmically generated persona system that is classified as a targeted technology. To test this technology, we conducted the study in a workplace environment. For the sample participants, we use real users. The sampling technique employed was random.*

As another example, a user study focusing on testing an interactive algorithmically-generated persona system using surrogate users could be reported as follows: *The study tests an algorithmically generated persona system that is classified as a targeted technology. To test this technology, we conducted the study in a laboratory environment. For the sample participants, we use surrogate users, who are crowdworkers. The sampling technique employed was convenience.*

An example of a generic study that always employs real users as participants could be reported as follows: *The study tests an algorithmically generated persona system that is classified as a generic technology. To test this technology, we conducted the study in a workplace environment. For the sample participants, we use real users. The sampling technique employed was convenience.*

## Limitations and future research avenues

The findings reported here involve some limitations. This systematic review is based on articles that self-identify as user studies and were published between 2013 and 2021. Our selection of user studies is skewed towards articles that explicitly refer to 'user study', even though some of the research in computing sciences may not do so. At the same time, 'user study' is an established conceptual phrase in IT, HCI, computing science, and related research when referring to testing technology with external stakeholders. Other selection methods for identifying user studies in IT could be pursued in future research. Additionally, one limitation of any archival database is the scope of the articles contained within it. The presented study could be replicated on other databases; however, the archival databases employed in this study were significantly large in both quantity and scope, and the articles employed were from 233 unique conference and journal outlets.

Another limitation concerns the interplay among user types, sampling techniques, and study validity. While *real users* are generally critical for most user studies, depending on the type and purpose of the study (*Hu, Chen & Wang, 2021*), there might be situations where the use of *surrogate users* would be appropriate, such as with pilot studies. Also, even if using *real users*, if the sampling technique is inappropriate, the external validity of the study might be questionable, as noted in the discussion of Fig. 3. Given the large number of user studies in our research that employed convenience sampling, this appears to be a concern. However, our research focused exclusively on the employment of *real* or *surrogate users* and did not examine the interplay with sample and validity or the use of control groups (*Li et al., 2022*). This is an interesting and challenging area for future work.

Moreover, follow-up research could address these five questions:

- *Who are the real users that user studies target?* Answering this question would provide a more nuanced understanding of the various types of *real users*.
- *What is the prevalence and impact of using students as participants?* Although students are often low-cost and convenient sub-populations, there are open questions concerning their use to represent *real users*, even for generic technologies.
- *Is the technology tested with an authentic use case?* This is another potential concern for validity, since testing technology in the circumstances of the real end-user is likely to result in more reliable findings.
- *Are there differences in the use of real users and surrogate users by publication venue?* This would lead to insights into the practices of researchers publishing in different venue types.
- *Who are the non-users of a technology?* The focus of most samples in the IT fields has been (when considered at all) identifying the users of that technology. A further avenue to explore may be identifying the non-users of a technology.

## CONCLUSIONS

Our analysis of 725 user studies shows that there is a potential systematic flaw in user studies for information technology regarding the sampling of *real users*, as nearly half of the user studies involving targeted technologies did not employ *real users*. Moreover, some IT fields have a higher prevalence of using *surrogate users*, especially students. Given the critical nature of *real users* in user studies, publication outlets should require an explicit statement in the article clarifying the type of users recruited for the research.

### Funding
The authors received no funding for this work.

### Competing Interests
The authors declare that they have no competing interests.

### Author Contributions
- Joni Salminen conceived and designed the experiments, performed the experiments, analyzed the data, prepared figures and/or tables, authored or reviewed drafts of the article, and approved the final draft.
- Soon-gyo Jung analyzed the data, authored or reviewed drafts of the article, and approved the final draft.
- Ahmed Kamel performed the experiments, authored or reviewed drafts of the article, and approved the final draft.
- Willemien Froneman performed the experiments, authored or reviewed drafts of the article, and approved the final draft.

- Bernard J. Jansen performed the experiments, analyzed the data, prepared figures and/or tables, authored or reviewed drafts of the article, and approved the final draft.

## Data Availability

This is a literature review; there is no raw data or code. The coding sheet with details for each article is available as a Supplemental File.

## Supplemental Information

Supplemental information for this article can be found online at http://dx.doi.org/10.7717/peerj-cs.1136#supplemental-information.

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
