# Peer review of "Who is in the sample? An analysis of real and surrogate users as participants in user study research in the information technology fields"

_PeerJ Computer Science, doi:10.7717/peerj-cs.1136_

## Round 0.1 · original submission · Major Revisions

Your submission was reviewed by two reviewers, whom I thank for their effort. Both reviewers welcome the scope of your work and have acknowledged several positives in the reporting style and methodological soundness.
They are, on the other hand, concerned by numerous aspects of the reporting, design, and validity of your study.

I invite you to submit a revised version and to address all comments of the reviewers in your response.

To be considered for publication, the submitted revision should place extra care on the following areas.

Both reviewers are concerned with the soundness of methodology that would be required by a systematic literature review, or more general meta-analysis studies. I understand that you have placed the submission to be what some call "survey of the literature" and that you have been clear in 179-180 that this is not a systematic literature review.

The submission, on the other hand, reports methodological details that go into that territory. Examples include the methodological description summarized in Figure 1, touching upon sample representation (of the collected papers, not of the users), and the quantitative part of the analysis. These are not negative points. On the contrary, they help to build the readers' confidence that the study was sound. These details, however, tend to frame readers in the direction of a systematic literature review.

I think that the submission could do a better job in framing what type of study was conducted. We, as readers, are at risk to start seeing the submission as a weak systematic literature review that does not want to be evaluated as one, rather than a sound survey of the literature. I prefer not to force a categorization of the submission, but proposals such as integrative literature reviews (see,. e.g., Torraco (2005)) or similar methods (see, e.g., Snyder (2019)) might help to frame the methodology better.

R1's concern regarding the single source for gathering papers (ACM Digital Library) is slightly mitigated if the submission becomes clearer that the study was not a systematic literature review, and it is re-framed under a specific umbrella of literature search. R1's concern is a valid one, nonetheless. Consider performing a round of backward and forward snowballing--see, e.g., Wohlin (2014)--from your dataset of included studies to widen the origin of your papers. This was also parts of the suggestions offered by R1.

The submission defines "surrogate" users and "real" users in lines 194-197 with the fundamental difference lying in the likelihood to be from/not from the target population that uses a technology under study. Both reviews touch upon the missed opportunity to consider issues of sampling and user characterization with respect to the--pardon my oversimplification--binary categorization of "in" or "out" of the interesting population.

R1 is concerned, in particular, with how the submission does not convince enough and glosses over the concept of sampling, which, I agree, is intertwined fundamentally with the interested populations. I invite the authors on elaborate on issues of sampling in their submission.

The words "surrogate" and "real" as modifiers to characterize users appear to be either not enough well-described / justified (R1) nor well-framed with respect to proxies for the intended population (R2) as in having students or researchers as participants to "surrogate" real users. Please provide better descriptions and rationale for the two terms and examples of sub-categories. The justifications provided in 199-204 and in 216-220 are not sufficient and bring a similar risk to the one that this is not a systematic literature review that appears to be one. I also agree with R2 that there appears to be an admitted fault in the provided definitions and categories, which confuses readers.

Moreover, please address R1's request for "decision rules or coding scheme guiding classification of real vs. surrogate" in-text.

As suggested by R2, elaborate more on M. J. Kim et al. (2013) given that you declare it as the only really highly related work. Please consider framing the results of your study with what is reported in M. J. Kim et al. (2013) as well.

Consider adding accompanying data, as suggested by R1, who provided concrete examples of the types of artifacts that could be shared without breaking any copyright statements.

Both reviewers have provided suggestions to widen your limitations section, and I recommend doing so.

I hope that you will agree to these suggestions, and I am looking forward to reading your revision.

Best regards,
Daniel Graziotin

References

Wohlin, C. (2014). Guidelines for snowballing in systematic literature studies and a replication in software engineering. In _Proceedings of the 18th international conference on evaluation and assessment in software engineering_ (pp. 1-10).
Torraco, R. J. (2005). Writing integrative literature reviews: Guidelines and examples. _Human resource development review_, _4_(3), 356-367.
Snyder, H. (2019). Literature review as a research methodology: An overview and guidelines. _Journal of business research_, _104_, 333-339.

·

Basic reporting

Good:
+ Well written with good grammar throughout
+ good use of figures and tables.
+ Includes PRISMA diagram
- Minor point: PRISMA diagram says “meta-analysis” but this study does not actually have any meta-analysis. Meta-analysis involves fitting a meta-analytic model to merge quantitative results.
+ key terms defined

Needs work:

There is no replication package in the materials I was provided. A replication package should be provided, including: complete search results, complete selection process results (organized into rounds if applicable); complete dataset (the spreadsheet, not the papers themselves), analysis scripts; coding schemes, examples of coding, decision rules (e.g. for classifying real vs. surrogate users), descriptions of edge cases.

The structured abstract usually has a "Method" heading. I also recommend using an "objective" heading.

The introduction would be more effective if it introduced the topic using more scientific language. This paper is about sampling. Sampling means selecting people or things from a target population. The problem with “surrogate” users is that we try to draw inferences about Population A from analyzing members of Population B. However, it is wrong to imply that a sample of users is representative by virtue of the users being “real”; i.e., members of the target population. A convenience sample of real users has no claim to representativeness. The whole paper would be more credible if discussed the issue at hand in the language of representative sampling, and acknowledged more clearly that selecting members of the target population is only the first step toward constructing a representative sample, and a very low bar indeed.

The next issue is that this paper combines a reasonably rigorous systematic review with an unreasonably polemical, one-sided essay admonishing researchers for using surrogate users. The “Why do real users matter?” and “Contribution to Research Community” sections unduly one-sided and lack a critical view of sampling issues. On the one hand, researchers should be allowed to state an opinion in (the discussion section of) their paper. On the other hand, the de facto thesis of this paper is “surrogate users are bad” and that thesis is not supported by the evidence or argument presented.

Different studies have different purposes. Lab-based experiments sacrifice external validity and realism and for internal validity and control. They use convenience samples, performing contrived tasks in artificial environments. Using surrogate users is just one more simplification that makes sense when maximizing external validity is not the aim of the study. In contrast, other kinds of studies (e.g. questionnaire surveys) sacrifice internal validity to improve external validity. If the point of the study is to maximize external validity, then, sure using surrogate users is unreasonable. However, using a non-representative convenience sample of real users may also be misleading. Anyone using convenience sampling complaining about someone else using surrogate users because the results won’t generalize is a massive hypocrite.

In summary, surrogate users are neither intrinsically reasonable or unreasonable; it all depends on the aims of the research. Now that’s my opinion. The authors are entitled to their opinions, but this is a systematic review, not an op-ed, so the whole essay about why surrogate users are bad needs to be cut way down and relegated to the discussion section.

Moreover, there are cases where surrogate users would be obviously preferable or superior. Building on the example on line 104, if my real users are 7-year-olds while my surrogate users are parents of 7-year-olds who happened to by HCI professionals or students, the surrogate users are likely to provide far more useful feedback. The paper should acknowledge such cases.

It simply isn’t necessary to take such a strong stand against surrogate users to motivate the study. The paper can just explain that statistically generalizing from study participants who aren’t actually in the target population to the target population is not valid. You don’t have to imply that all studies aim for such generalization.

Experimental design

Good
+ specifies the data extracted from each primary study
+ acknowledges potential blurry distinction between real and surrogate users
+ clear chain of evidence from the extracted data to the answers to the research question(s)

+/- having a second researcher audit the first researcher’s coding is better than nothing, but not as good as two researchers doing the coding independently and quantitatively assessing reliability.

Needs work:
- Only searched one database, the ACMDL. This is a serious problem.
- Searching keywords, specifically, is not ideal because use of keywords is unpredictable
- No explicit screening criteria
- no steps to mitigate response bias (e.g. multiple database, google search, reference snowballing)
- does not state the dates on which the search occurred
- does not provide decision rules or coding scheme guiding classification of real vs. surrogate
- Methods section does not follow typical organization/subsections of a systematic review (e.g. search strategy, search terms, selection criteria, screening procedure, data extraction, disagreement resolution, data analysis)

The search strategy creates substantial sampling bias, which undermines the external validity of the study.

The biggest problem here is only searching keywords on a single database. The ACMDL is not the only relevant source of papers performing user studies, and there’s no justification for limiting data collection to ACMDL. The authors should have also searched Scopus, SpringerLink and IEEE Xplore. Repeating the authors’ search process on IEEE Xplore turned up 700 papers.

I understand the issue with getting more articles than you can manually screen. However, the solution is to iterate on the search process, adding “not’s” to the search string to remove extraneous results. If it’s still too much, then you download all of the records, de-duplicate, and use probability sampling to select a random subsample for the study. That’s defensible. Searching keywords instead of abstracts an only using one database is not a defensible sampling strategy.

The screening process would have higher validity if two different researchers did some or all of the screening independently, and statistically analyzed inter-rater reliability (e.g. using Krippendorff’s Alpha). This is not strictly necessary, but it is better.

251-2: “When there is a mix of participants, we choose the largest group as the representative group” First, that’s not what representative means. You mean, “When there was a mix of participants, we classified the study based on the largest group”. Second, however, this is a serious limitation of the study. It is replicable, but the results would be more credible if mixed-groups had been broken down somehow. How many groups were mixed? If it’s a small percentage, it can just be noted as a limitation. If it’s a large percentage, I think you should find a better way of dealing with it.

374: If you are going to focus on “targeted technologies” then that should have been a screening criterion, and all the papers evaluating generic technologies should have been excluded from the beginning. In this study, n=256, not 571.

405 / 476: The use of logistic and poisson regression here seems like just an excuse to get some p-values in the paper. This paper neither tests a priori hypotheses nor builds a theoretical model. Applying inferential statistics therefore seems superfluous. Simple bar charts showing the percentage of studies in each category that have real users would be more appropriate. Use the simplest methods that do the job.

The limitations section needs work. It should explicitly acknowledge potential sampling bias caused by database selection, search string and operationalization focused on keywords, all of which threaten external validity. Meanwhile, use of a single rater threatens reliability. Use of audits partially mitigates this threat. Regarding internal validity, the paper claims that using “real” users will lead to more representative sampling and more useful results, but no evidence supporting this claim is presented. If this theme is downplayed, there would be no need to address it in the limitations.

Validity of the findings

87-88: “As the results of user studies are often evaluated statistically, the sampled participants should represent the characteristics of the underlying population” - Statistical evaluation does not make a sample representative; the sampling strategy is what determines representativeness.
91: Real users may also be unmotivated
93: testing an app for the visually impaired on normally-sighted people wearing eyeglasses that simulate a specific visual impairment might work just fine.
98: the quintessential question underlying scientific validity is “are the results TRUE”.
115: talking to surrogate users does not necessarily lead to toy problems.

140: where the users come from is irrelevant to “valid statistical analysis.” External validity is not the same as conclusion validity.

Table 1 is not helpful. (other tables are good)

180: “Our sampling is better understood as a form of sampling” - what does this mean?
181: if the current study is not a systematic literature review, what is it? A critical review?
184: You didn’t “manually validate the sample.” This step is called “screening”.
204: This would be more convincing with discussions of edge cases or some example classifications

244-249: Above, the paper notes that students could be real users depending on the product, but here students are treated as surrogate users. How were cases coded where the users were real users and surrogates? (This issue is revisited on lines 275-277 but it’s still murky - please clarify).

332-335: this doesn’t seem to belong here.

344-349: Shouldn’t this appear two pages earlier where these categories are explained?

524: the IRB is an American concept. Each country is different. Don’t misapply American practices to global research communities.

The conclusion should do a better job of summarizing the paper’s contributions.

Additional comments

To summarize, I recommend inviting a revision. The authors should address all of my comments, but the main things are:

1) Re-do the data collection using a better search strategy and more databases
2) Provide a comprehensive replication package
3) Tone down the tirade against surrogate users; present a more balanced view grounded in the language and logic of representative sampling
4) Reorganize the method section into common subheadings used in systematic reviews

Sorry it's a lot of work, but I'm comprehensive on the first round so there's no surprises in the second round.

·

Basic reporting

The review is of broad and cross-disciplinary interest and within the scope of the journal, and conforms to the guidance for literature review articles.

The introduction part did a good job to setup the context of the study and motivate the it by arguing the significance of the topic and the knowledge gap drawing up relevant literature. Even though there were several literature reviews of the user studies, they were not as extensive as this one and only one addressed use types employed.

The English writing is simply and concise, therefore, makes an easy and pleasant reading experience for me.

Experimental design

This article content is within the aims and scope of the journal. The overall research design is thoughtful, rigorous and justified in a convincing manner. No ethical concern was raised. The sampling of the papers and user studies and the extraction and analysis of data were described with sufficient detail, which ensures its replicability.

However, I do observe a set of issues related to data analysis and results reporting. I report them and suggestions to improvement in the next section, together with the points in that section.

Validity of the findings

The novelty of the research findings were argued in the Discussion section, but not sufficiently grounded in the in-depth review of related work (e.g., regarding Kim et al. 2013), and not organised according to and linked to the three RQs.

I really liked the template the author provided for reporting the role of users. I can see its usefulness and value to improve the quality of reporting research. It can be also useful to guide the design of a study.

Limitations and future work are reflected upon.

Following are observations/suggestions regarding the study design the validity of the findings, following the order of importance:

1) see Line 246-247, the categorisation of user types is somehow misleading, since students, crowd workers or researchers can be real users, as shown in the results later (Line 346-349). Therefore, they are not always surrogate users. It is better to have two top categories: real users and surrogate users, then within each category, there can be sub-categories in which students, crowd workers or researchers can be sub-categories. Following this, the statement in Line “248” is problematic, as a user type “students” is not necessarily surrogate user, it can be real user depending on the study. Another point to clarify: are these categories predefined, or emergent from data? It is not clear in the text.
Related to this comment, Line 601-605, the provided template, I think the part related to user types could be updated as “… we use [REAL USERS/SURROGATE USERS (CROWD WORKERS/STUDENTS/RESEARCHERS]…”

2) Technology types and domain types are not clear distinguished. What’s true difference? The section “Technology and Domain Types” (Line 279-330) described them in a mixed manner without making a clear separation. In addition, the content in this section is already about the analysis results. Maybe better to move the content to the Results section, and report the domain types and technology types more systematically, rather than just picking up some categories without explaining the rationale of this picking. A related comment: “Data Processing” (Line 373-389) is more about data analysis than results, therefore should be moved to “Survey Methodology” section? In general, it would be nice to streamline the text in the Results section following the three research questions.

3) see Line 154-156, “… only one of the user study reviews …” It would be nice to say more about this review, as it seems the most relevant work to this study. What are the key findings from this review? In the Discussion section, the findings of this study should be contrasted to this relevant review, to highlight further the original findings from this study. A more general comment on the Discussion section: it would be nice to organise the discussion according to the three research questions, to provide more insights on the findings related to these research questions drawing upon relevant literature.

Additional comments

This paper is a meta-analysis of a selection of user studies conducted in Computing Science research fields, to understand how real users vs surrogate users were used in IT user studies. 571 user studies were sampled and analysed to answer three research questions: how often do IT user studies use real users, who are surrogate users, and whether the use of real users vary by research domain and technology studied. The authors conclude that most user studies in Computing Science employ real users, even though the extent varies depending on research domain and technology type. Nonetheless, the authors highlight the need to report explicitly the type of users employed, and provide a template to report the role of users in user studies.

The authors addressed a quite interesting and meaningful aspect of user studies – the types of users employed in terms of real vs. surrogate. As the authors noted in the paper, this is a somehow overlooked aspect and many studies don’t explicitly declare the types of users used. Using a template to provide this information could help the readers better evaluate the research findings in terms of external validity and usefulness to practice.

The large number of user studies analysed, as well as the mixed qualitative and quantitative analysis on the collected studies, are another strength of this submission.

Following is a set of minor comments/suggestions that I wish could further help the authors to improve their study and the paper. These points are given following the sequence of the sections.

- Line 78, in the definition of user study, the term “system” is worth further clarification. If I understood correctly, it is not limited to IT systems, but can be e.g., software technique or practice as well.

- Line 106, “Problem 3” should be “P3”, to be consistent with the previous “P1” and “P2”. Is this “vicious cycle” related to “applicability”? If yes, I would suggest to call P3 “applicability”, to be consistent with the text in, e.g., Line 142.

- Line 134, “students’ excessive use” is misleading, maybe better “the excessive use of students”?

- Line 150-151, “… the focus of past surveys is most often on methods, findings, and sources of data.”, would be “users studied” sources of data?

- Line 165, please provide the reference to PRISMA, and maybe a justification of why using this guideline if this study is NOT a systematic literature review.

- Line 171, “we searched the ACM … with the phrase ‘user study’ …”, please clarify whether all fields were searched, or only “title” field was searched?

- Line 332, please explain how “89.6%” is calculated?

- Line 397-398, “We surmise that surrogate users in the information processing … are related to the study of mainly generic technologies.” I don’t understand this statement. Users employed in the user studies of generic technologies are not always considered real users, so no surrogate users in the studies of generic technologies?

- Line 437, “5.32”, not “5.31”, according to Table 9.

- Line 448, “The results in Table 10 show that …”, it should be “Figure 3” rather than Table 10 to be referred to here.

- Line 590, “Note that the problem is not related to validity…”, it should be “… is not ONLY related to validity…”?

- Line 612, another piece of future work could be analysing real users vs. surrogate users by publication venues, e.g., conferences. vs. journals, or difference conference or journal venues.

- Table 1, I am not sure it works better than plain text.

- Table 2, He&King is missing from the “References” list.

- Table 3 is a mix of information on the articles (n=489) and identified user studies (n=571). It may make sense to have two tables, or make it clearer that Table 3 is not about articles, but about the user studies.

- Table 6, please spell out the acronyms of the venues for those who are not familiar with them, and indicate which is conference and which is journal. Another point: why isn’t the total of the percentage not 100%?

- Table 7 is somehow redundant. The data is completely covered by Table 4.

- Table 8, it would be nice to show the mapping between the originally identified domains (D1-D19 in Table 4) and these superclasses, to show the traceability of data analysis.

- Figures 2 and 3, why is the blue legend labelled as “other users”, not “surrogate users”?

- Figure 4, it is very vague the terms used in the diagram. There is no explanation how “highly preferable”, “Preferable”, “Possible” are determined for these quadrants?

---

## Round 0.2 · Major Revisions

Dear authors, thank you for submitting your revision. 

I agree with both reviewers that the revision significantly improves on the initial submission.

Reviewer 2 has observed and summarized many of the issues that I was anticipating when I reached my first editorial decision. If this submission has to be evaluated as a systematic literature review, there are important steps to be followed and details to be reported. The paper can, and should, improve on these aspects.

In particular, Reviewer 2 indicates four concerning points on missing methodological explanations (search query, snowballing, inclusion criteria, PRISMA). I see addressing these four concerns to be instrumental in considering publication of the submission.

My initial suggestion was to frame the study following proposed designs (e.g., by Snyder) and avoid the terminology "systematic literature review". I will not impose on your decision to go the opposite way. It is, however, apparent that the study was not designed to be a systematic literature review in the first place. Kindly note that this would not imply a lesser quality of the study or value in its results. On the contrary, a survey of the literature that borrows from the systematic literature review methodology is valuable, transparent, and enhances how readers trust its results. 
Hence, my proposal is to address Reviewer 2's concerns and make sure that mentions of "systematic (literature) review" in the manuscript become accompanies by characterizations such as "inspired from" and "borrowed from". I hope that the authors will agree with me that such a compromise would inspire intellectual honesty and transparency, to the readers. 

Reviewer 1 has left a wholehearted suggestion that I recommend considering. I agree with Reviewer 1 that the manuscript could do a better job in keeping its neutrality on using surrogate users. I leave the decision to follow the recommendation or not, to you, but I ask you to address it in a response letter at the least.

The reviewers have offered other suggestions, including minor comments, that I invite you to observe, but the above ones form the core of my editorial decision.

I will also add the following minor comment:

Abstract, 36-37: "and the remaining sampled 6.7 researchers or did not say" is unclear in what it means. The remaining 6.7% studies sampled researchers or did not say?

I am looking forward to reading your revised manuscript.

Kind regards,
Daniel Graziotin

·

Basic reporting

The authors have done a good job addressing my comments. No further concerns here.

Experimental design

The authors have done a good job addressing my comments. I appreciate the extra work done on inter-rater reliability and including additional databases. No further comments here.

Validity of the findings

I'm happy with the changes other than two small things:

Fig 3: again, while using surrogate users obviously threatens external validity, using real users doesn’t guarantee high external validity.

line 135: “a systematic review is conducted”
line 297: “although our research is not a systematic review…”

Additional comments

I'd like to offer the authors some friendly advice. Although I see where the authors have moderated their positions, this paper comes out very strongly against using surrogate users. Many of the potential readers of this paper have used surrogate users in studies. They may feel attacked by the paper. When people feel attacked, they stop paying attention, and the paper won't affect their opinions or behaviour.

If you want to have more impact - if you want to persuade - empathize with the reader. Show that you understand why researchers recruit surrogate users.

A between-subjects randomized controlled trial to identify a modest effect size requires ~200 participants. In my country, getting 200 professional programmers to spend a couple of hours in a lab will cost something like $60,000. No funder in North American will hand out $60k just for incentives for one study. Similarly, to ethically justify recruiting visually impaired people to participate in a study, we’d have to pay them for their time and possibly arrange transportation to a test site. This is expensive. Meanwhile, we can get students to participate in research for free, for the learning experience, or for marks (with various mechanisms so they’re not coerced).

If you want to persuade researchers to use more real users, give actionable suggestions. How can I get 200 software professionals into my lab for a few hours without paying them? There was a workshop at ICSE this year, ROPES, where this was discussed at some length. No one really had an answer. If you can answer this question, the paper could have a lot of impact.

If you can't answer this question, don't be so critical.

·

Basic reporting

The original review on this aspect still holds.

Experimental design

Please see the additional comments part.

Validity of the findings

The validity of the findings was improved. Please see the additional comments part for some specific concerns.

Additional comments

I appreciated that the authors took the critical comments from both reviewers in a positive manner. The revision showed the major effort from the authors. The responses to the comments are mostly thoughtful, with a good level of detail. The highlights in the revised word document eased the review process.

My major comments were addressed by the authors, even though I still have some concerns (elaborated below). I believe that, through this major revision, the manuscript is in a better shape, and with improved clarity.

I wish the following comments, some on the responses, some on the revised text, could help the authors to further improve their study.

1. In the responses to reviewers, the authors made it clear that they now consider the study is a systematic literature review. However, with this explicit claim, the authors also set up an explicit expectation in a reader’s mind (at least in my mind) to see a more rigorously executed review process to qualify it as “systematic”. That means, every decision made in each step should be justified.

1) I am not convinced by the fact that the search in the two databases (ACMDL and IEEE Xplore) was not done in the same manner: “Keyword” field was searched in ACMDL and “Title” field was searched in IEEE Xplore. Why not using the same field across the two fields?

2) What is this modified snowballing approach? Why snowballing in WoS, and why identifying highly cited user study articles? Is this what snowballing mean?

3) What are the inclusion/exclusion criteria used in the manual screening step? The only criterion seems hidden in Table 2: “Actual user study” (which shouldn’t be in the table as a data extraction attribute).
4) The authors claimed that they followed PRISMA. It is not very clear to me how they followed it rigorously to make the whole review process “systematic”.

2. The authors did a good job in the Introduction section to motivate the significance of investigating the use of real users in user studies. However, I felt that the author somehow overdid it by dedicating too much space to it (Lines 147-292). Lines 147-260 contained some repetitive information. E.g., are Lines 241-260 really necessary? The arguments of the research gap starts only from Line 262. The authors can be much more parsimonious with the text here.

3. I still think the authors could make a better decision on what content goes into a methodology section, and what goes into the result section. Some content presented in the methodology section, including the four groups of surrogate user (Lines 400-408), the 19 types of IT fields (Lines 459-477) are results achieved, not methodological choices in my opinion.

4. A list of minor comments, Following the order of the sections:

- In the abstract, Line 27, remove the first “published”? Line 38, the text is about “Conclusions”, or “key findings”? read more like “Results”

- Line 66, “… and can use such techniques as a sample frame…”, maybe it makes more sense like this: “… and can use techniques such as a sample frame…”?

- Line 297: Is your research a systematic review or not?

- Line 347, please check the math.

- Lines 368-370, this is not a good example of ‘the distinction between “real” and “surrogate” is not always clear and distinct’. The two examples given here are actually quite easy to determine whether the student participants are real or surrogate users. I would drop this example. The other two examples of gaming and mobile technology are fine in my opinion.

- Line 422, logically, the subsection “Type of User Study” should go before “Type of Users Employed”, since the determination of type of users depends on type of user study.

- Line 424, “...(see Table 2),”, not Table 1.

- Lines 795-806, the same limitation was described several times in the same paragraph. This paragraph can be much more compact.

- Lines 811-812, well said, however, it is somehow contradicting to Figure 3, which gives the impression that using real users would always brings high external validity!

- Figure 1, it is strange to read “Records after duplicates removed (n=14242)”, does it mean there is no duplicate in the search results from the two databases, and from the other sources?

- Figure 3, the text under the title was incomplete.

---

## Round 0.3 · Minor Revisions

Dear authors,

Thank you again for your revised manuscript and for the extensive responses to the reviewers.

In my opinion, which is shared by both reviewers now, we are close to reaching a state where the paper provides enough details and clarifications that bring trust in evaluating the study soundness.

I agree with the reviewer that the use of snowballing could be better explained. We have the following mentions of snowballing:

- 141-142 these articles are drawn from two major digital libraries and supplemented with snowball sampling (Wohlin, 2014) from a third digital library
- 249-252 To address our research questions, a systematic search using the search phrase 'user study' was conducted on the major information technology databases ACM Digital Library and IEEE Xplore, with snowball sampling on Web of Science for articles published from 1 January 2013 to 31 December 2021
- 297-302 Finally, using a modified snowballing approach, we identified thirteen highly cited user study articles in Web of Science (WoS) from various journal outlets using the highly cited feature in the WoS interface and the search phrase 'user study'. Snowballing using citations of references in identified articles is a common technique in systemic reviews. Our approach of using highly cited articles from the WoS is similar and possibly more representative and less biased than the traditional method of picking cited articles from the reference list of articles in the dataset.
- 304-305 Our focused search criteria and focused snowballing resulted in no duplicates needing to be removed. 

-

I ask you to:

1. Streamline terminology (snowball sampling, modified snowballing, focused snowballing) for all mentions.
2. Elaborate on what deviated from snowballing as introduced by Wohlin (2014).
* In short, snowballing is about having a starting set of papers and (a) go back through their reference list and (b) go forward through new papers that cited the current ones. What did you perform that is different from (a) and (b)?
* The text seems to suggest that you performed (a) and (b) but limited to 2013-2021 (251-252) but also that, at the same time, you only sampled some highly cited papers from the WoS (297-299). I am unsure how these papers were identified from a snowballing session.
3. Clarify, in text, how this approach would be more representative and less biased.

I am looking forward to reading your revised manuscript.

·

Basic reporting

Again improved in comparison to the previous version.

Experimental design

More details are provided to render the study design more systematic. Please see the additional comments part for some additional concerns.

Validity of the findings

The previous evaluation on this aspect still holds. Overall positive. Please see the additional comments part for some additional concerns.

Additional comments

Thanks to the authors for another round of careful review of their manuscript. Again I appreciated that the authors engaged the reviewers in a positive and constructive exchange of thoughts. I am happy with how the authors addressed my comments and how the manuscript was revised. I believe that it is close to be publishable after a final minor adjustment.

Below are some remaining doubts and minor comments which I wish will help towards this end:

- I appreciated the recommended techniques to recruit real users (Lines 665-688). However, the authors could provide more balanced view on these techniques, e.g., wha could be the potential pitfalls of using crowdworker platforms? Why emphasizing the commercial nature of the partner companies? Would the commercial nature bring in economic concerns and therefore bias when recruiting real users?
By the way, isn’t your recommendation of using crowdworker platforms in contradiction to the “desk rejection” example you cited in the introduction (Lines 93-96)?
A minor comment: Line 683, “… a larger SAMPLE...”

- Line 289, “by IEEE and full text.” What does this phrase mean here?

- Regarding the response to the modified snowballing approach, I am still not convinced it could be called as such. This modified approach doesn’t convey the meaning “snowballing”, it is one sampling method used by the authors to select the studies from WoS, not snowballing from a set of papers. Also, the claim made in Lines 300-302, “Our approach … is … possibly more representative and less biased …” is not convincing to me. Why is sampling highly cited papers “more representative and less biased” than traditional snowballing?

- Lines 796-802 may need another careful check. The limitation of only “keywords” field was searched holds true to ACMDL, but not to IEEE Xplore and WoS. Any limitations regarding these two databases?

---

## Round 0.4 · accepted · Accept

Thank you for addressing all concerns and for your cooperation.

I believe that the manuscript, in the present state, provides solid transparency about a sound methodology. And an interesting, insightful, research endeavour.

My personal thanks to the reviewers, as well, for providing such elaborated, constructive feedback.